# Frequent Lucid Dreaming Is Associated with Meditation Practice Styles, Meta-Awareness, and Trait Mindfulness

**DOI:** 10.3390/brainsci14050496

**Published:** 2024-05-14

**Authors:** Elena Gerhardt, Benjamin Baird

**Affiliations:** 1Institute of Psychology, Osnabrück University, 49076 Osnabrück, Germany; egerhardt@uni-osnabrueck.de; 2Department of Psychology, The University of Texas at Austin, Austin, TX 78712, USA

**Keywords:** lucid dreaming frequency, mindfulness, meditation practices, meta-awareness

## Abstract

Lucid dreaming involves becoming aware that one’s current experience is a dream, which has similarities with the notion of mindfulness—becoming aware of moment-to-moment changes in experience. Additionally, meta-awareness, the ability to explicitly notice the current content of one’s own mental state, has also been proposed to play an important role both in lucid dreaming and mindfulness meditation practices. However, research has shown conflicting strengths of associations between mindfulness, meditation, and lucid dreaming frequency, and the link between lucid dreaming and meta-awareness has not yet been empirically studied. This study evaluated the associations between lucid dreaming frequency and different meditation practice styles, mindfulness traits, and individual differences in meta-awareness through an online survey (*n* = 635). The results suggest that daily frequent meditators experience more lucid dreams than non-frequent meditators. However, weekly frequent meditators did not have a higher lucid dreaming frequency. A positive association was observed between open monitoring styles of meditation and lucid dreaming. The findings also indicate that meta-awareness is higher for meditators and weekly lucid dreamers. Furthermore, frequent lucid dreaming was commonly associated with a non-reactive stance and experiencing transcendence. Overall, the findings suggest a positive relationship between specific meditation practices and lucid dreaming as well as the importance of meta-awareness as a cognitive process linking meditation, mindfulness, and lucid dreaming.

## 1. Introduction

Being explicitly aware of one’s own mental state and maintaining present-centered awareness—paying attention to moment-to-moment changes in thoughts, emotions, and perceptions with a non-judgmental stance—are essential elements of the definition of mindfulness [1,2]. Becoming aware that one is dreaming while still being asleep defines the nocturnal state of lucid dreaming [3,4]. In both mindful awareness and lucid dreaming, there is an explicit awareness of one’s current mental state that characterizes meta-awareness [5,6]. This awareness can be either propositional (“I am dreaming” or “I am mind-wandering”) or non-propositional, sustaining a peripheral awareness of engagement with a chosen object or the ongoing realization of being in a dream [7]. Mindfulness can be cultivated through the practice of meditation, while the ability to induce lucid dreams can be trained through various methods, including cognitive practices, induction devices, or the use of substances [8,9,10].

Scientific research has focused primarily on meditation practices within Buddhism, which is divided into numerous lineages [11]. In Tibetan Buddhism, the achievement of continuous conscious awareness during all stages of sleep and dreaming is highly valued [12,13]. Tibetan “Dream and Sleep Yoga” teachings, a set of daytime and nighttime practices for gaining awareness during dreams and using lucid dreams as a platform for various meditation practices, were reserved for advanced practitioners [14,15]. Lucid dreaming and dreamless sleep awareness are seen as pathways to spiritual growth and enlightenment [14,15]. Meditation consists of deliberate actions such as observing, concentrating, letting go, generating, visualizing, and shifting attention from one mental object to another, all anchored in conscious awareness. Central to this practice is meta-awareness that integrates these various activities. Such a practice is diversely implemented across cultural traditions and secular settings [16].

Meditative practices may be differentiated depending on the direction and dynamics of attention. For instance, focused attention (FA) meditation involves practices that narrow the scope of attention and cultivate one-pointed concentration on a single object, such as the observation of the breath. Open monitoring (OM) meditation involves the non-reactive monitoring of the content of experience from moment to moment [17]. FA meditation requires the stability of attention on an object or activity, whereas OM meditation is based on the openness and expansiveness of awareness, monitoring changes over time [18]. Hence, meditation practices cultivate specific mental skills, including enhanced attentional stability and the monitoring of one’s mental state (i.e., meta-awareness) [18]. Meta-awareness is an essential cognitive capacity discussed as the primary cognitive component for the state shift of non-lucid to lucid dreams [19]. Non-propositional meta-awareness is considered particularly relevant to lucid dreaming [7]. Meta-awareness becomes crucial at the moment of becoming aware of the bizarreness of the dream plot, recognizing the dream signs, or passively observing the ongoing dream [20,21,22,23]. Even in a stable lucid dream, one has a sustained awareness of the dream state while experiencing events in the dream [7,24].

Building on the idea that not just the waking state offers self-reflective thought and cognition but that self-reflective awareness can also be achieved in the dream state, a theoretical consideration of connecting lucidity with mindfulness influenced by meditation was made, placing waking experiences, dreams, and several stages of lucidity on a continuum of self-reflectiveness [25]. Taking this idea further, enhancing mindfulness during the day is thought to also increase mindfulness during the night [26]. This is derived from the continuity hypothesis that waking memories, dispositions, and habits are incorporated and transferred into the dream state [27]. 

Despite strong theoretical linkages, the degree of associations between meditation, mindfulness, and lucid dreaming have been conflicting in empirical studies. Overall, findings suggest an increase in the frequency of spontaneous lucid dreams in frequent or long-term meditators [28,29,30,31,32,33,34]. Moreover, evidence suggests that the extent of meditation experience can alter the strength of the association between mindfulness and lucid dreaming [33]. One recent investigation did not report an increase in lucid dreaming frequency due to meditation experience. However, it did find that certain facets of mindfulness were positively correlated with lucid dreaming frequency [35]. In a recent study by Geise and Smith [36], the Transcendence subscale of the Relaxation, Mindfulness, and Meditation Experience Tracker was found to be a significant predictor of lucid dreaming frequency. However, the total score on the Freiburg Mindfulness Inventory, both subscales, Presence and Acceptance, and measures of meditation frequency or experience did not show significant correlations with lucid dreaming frequency [36]. A positive association between an estimate of the number of lucid dreams during one year and the total number of years of meditation experience has also been observed [31].

Gackenbach et al. [29] found that intensive and frequent meditators rooted in Transcendental Meditation, on average, experience lucid dreams once or more per week. Despite the methodological issues regarding the potentially biased selection of participants for the study, the evidence for higher lucid dreaming frequencies in populations of long-term and frequent meditators should be considered. Previous findings from Baird et al. [28] indicated that long-term meditators have more spontaneous lucid dreams compared to inexperienced meditators. Furthermore, the associations between aspects of trait mindfulness measured by the Five Facet Mindfulness Questionnaire, such as Acting with Awareness and Observing, and the Decentering subscale of the Toronto Mindfulness Scale, were higher in long-term meditators with frequent lucid dreams. However, meditation novices did not show an increase in lucid dreaming frequency after 8-week mindfulness meditation training. A recent study validated an indirect lucid dream experience questionnaire in Spanish and examined, among other dream-related constructs, meditation practices, experience, and aspects of mindfulness [34]. The results suggested that time spent in OM meditation was positively correlated with higher scores on the lucid dream aspects of insight and control. The study did not examine lucid dreaming frequency, nor did it validate whether participants ever had a lucid dream.

These results are in line with the interpretation that meditation training improves metacognitive skills with the enhancement of dispositional mindfulness, which in turn could increase nighttime meta-awareness in order to promote the state shift of the onset of lucid dreaming. Research shows that different meditation techniques within various frameworks and traditions have different effects [37,38]. Furthermore, there has not been any study investigating which meditation technique is associated with higher lucid dreaming frequency. This study therefore explored the connection between meditation practices, meditation frequency, dispositional mindfulness, and lucid dreaming frequency. First, we sought to replicate the empirical findings that frequent meditators exhibit higher lucid dreaming frequencies. Second, we evaluated individual differences between frequent meditators and non-meditators on all mindfulness facets and dreaming variables to further explore the relationship between trait mindfulness and lucid dreaming. Third, we studied the relationship between lucid dreaming frequency and specific meditation techniques/practice styles. The main hypothesis was that particularly open monitoring (OM) meditation, and possibly focused attention (FA) meditation, would have a positive association with lucid dreaming frequency, as both practices emphasize the cultivation of meta-awareness and sustained attention monitoring. Moreover, it was expected that meta-awareness would be higher in frequent meditators compared to non-frequent meditators, but also in weekly lucid dreamers compared to non-weekly lucid dreamers. Lastly, we explored the role of meta-awareness in the association of meditation frequency and lucid dreaming frequency. 

## 2. Methods

### 2.1. Participants

In total, 635 participants completed the online survey. Only persons who met the following criteria were asked to complete the survey: all participants (1) must be at least 18 years and no more than 75 years old, and (2) must be fluent in English. The upper limit of age was set, as lucid dream incidences and cognitive capacities have been shown to decline over age [39]. The convenience sample splits up into a German student population from Osnabrück University (Uos, *n* = 72) and students from the University of Texas at Austin (UT Austin, *n* = 272), as well as a general mixed international sample of 291 respondents. The study was approved by the Institutional Review Board of the University of Texas at Austin (STUDY00003582). Notably, all participants who claimed to have experienced at least one lucid dream had to verify their understanding of the lucid dream experience. Participants who did not pass the verification were not eligible and were thus excluded. All respondents were grouped depending on their meditation frequency [40,41]. Participants were either classified as non-frequent meditators (i.e., meditating less than once per week), as weekly frequent meditators (WFMs; meditating once or more per week), or as daily frequent meditators (DFMs; meditating at least twice per day or multiple times daily).

Within the Osnabrück University sample, 66 participants were eligible (49 females, 16 males, and 1 non-binary; age = 22.94 ± 5.6 (M ± SD)). Based on meditation frequency, 2 students were classified as DFMs, 11 meditated weekly, and 53 were non-frequent meditators. Within the student population from UT Austin, 241 were classified as eligible (156 female, 81 male, and 4 non-binary; age = 19.43 ± 1.89 (M ± SD)). Based on meditation frequency, 1 student was classified as DFM, 29 meditated weekly, and 211 were non-frequent meditators. Within the general mixed population, out of 291 initial respondents, 270 were eligible, more male respondents completed the survey (112 females, 149 males, 4 non-binary, and 5 self-described, e.g., “Genderfluid”), and their ages ranged from 18 to 75, with an average of 37.74 ± 16.16 (M ± SD). Within the general mixed sample, 35 participants were DFMs, 117 were WFMs, and 118 were classified as non-frequent meditators. 

### 2.2. Procedure

The survey was internationally distributed, with a focus on Europe and the United States. Recruitment started in January 2023 and ended in July 2023. Data collection and the entire recruitment process were conducted online. All study materials were provided in English and implemented using Qualtrics software (https://www.qualtrics.com) (accessed on 1 January 2023). The survey completion took, on average, 54.8 (Mdn) minutes in the general sample, while the student populations needed between 20.9 (UT Austin; Mdn) and 35.7 (Uos; Mdn) minutes. The platforms for distributing the survey varied. The link and an invitation protocol were sent out to several lucid dreaming experts, institutes, and other researchers. Additionally, placement on several social media platforms, starting with Reddit, Facebook, Instagram, Forums, and other websites like Dream Views, YouTube, and LinkedIn, achieved a wide reach. Furthermore, meditation and wellness centers facilitated study participant recruitment by distributing materials through email lists.

Students from UT Austin were reached via the SONA system. Psychology and Cognitive Science students from Osnabrück University were contacted via mailing lists. All student populations were compensated with one credit, which corresponds to an hour of participation, for their research participation sheets. Respondents in the mixed general sample did not receive any compensation. All participants were provided with the informed consent document and a short introduction. After the agreement, all participants received the following sections in the same order: Demographics, Dream Survey, and Meditation Experience Questionnaire. For these instruments, branching allowed the researchers to efficiently present participants with in-depth questions based on their previous experience. The following instruments were presented to all in a randomized order: Multidimensional Awareness Scale, Toronto-Mindfulness Scale, Relaxation, Meditation, and Mindfulness Experiences Questionnaire, and Five Facet Mindfulness Questionnaire, along with two other scales which were part of another research scope (Mysticism Scale and Indirect Realism Scale).

### 2.3. Measures 

In order to achieve a comprehensive assessment while maintaining time efficiency, shortened versions of many instruments were implemented. For internal consistency of the scales, McDonald’s omega total was preferred over Cronbach’s alpha. Since it can be assumed that not all items contribute equally to their score, McDonald’s omega is a more accurate reliability estimate, especially for multidimensional or ordinal scales [42,43,44]. Omega can be described as the proportion of variance in observed scores that can be attributed to a single underlying factor or to the common variance among the items on a scale [45,46]. As with Cronbach’s alpha, larger values indicate a higher reliability [47]. 

Dream Recall and Lucid Dreaming Experience. Lucid and ordinary dream experiences were recorded with an adapted dream survey. The original questionnaire developed by Baird et al. [48] was modified to fit the specific aims of this study. All participants reported dream recall frequency and lucid dreaming frequency on a 16-point Likert scale, extending the established scales by Schredl and Erlacher [49]: 0 = *never*; 1 = *less than 1 (lucid) dream per year*; 2 = *1 (lucid) dream per year*; 3 = *2 (lucid) dreams per year*; 4 = *3–5 (lucid) dreams per year*; 5 = *6–8 (lucid) dreams per year*; 6 = *9–11 (lucid) dreams per year*; 7 = *1 (lucid) dream per month*; 8 = *2 (lucid) dreams per month*; 9 = *3 (lucid) dreams per month*; 10 = *1 (lucid) dream per week*; 11 = *2 (lucid) dreams per week*; 12 = *3–4 (lucid) dreams per week*; 13 = *5–6 (lucid) dreams per week*; 14 = *1 (lucid) dream per night*; 15 = *more than 1 (lucid) dream per night*. Based on the methodology of Stumbrys, Erlacher, and Malinowski [33], class means transformed the ordinal scores into metric frequencies either as units per month (0 → 0, 1 → 0.0714, 2 → 0.0833, 3 → 0.1667, 4 → 0.3333, 5 → 0.5833, 6 → 0.8333, 7 → 1, 8 → 2, 9 → 3, 10 → 4, 11 → 8, 12 → 13.5, 13 → 23.5, 14 → 30, 15 → 33) or units per week (0 → 0, 1 → 0.0185, 2 → 0.0192, 3 → 0.0385, 4 → 0.0769, 5 → 0.1346, 6 → 0.1923, 7 → 0.25, 8 → 0.50, 9 → 0.75, 10 → 1, 11 → 2, 12 → 3.5, 13 → 5.5, 14 → 7, 15 → 9). The same class means-recoded 16-point scale was given for lucid dream induction frequency per month. Participants received a written definition along with the scales: “Lucid dreaming is a special sort of dream in which you know that you are dreaming while still in the dream. Typically, you tell yourself “I’m dreaming!” or “This is a dream!”. In some cases, you may also control the content of the dream and alter the dream events as well as your actions voluntarily”.

Respondents who had previously experienced lucid dreams were asked detailed questions regarding their lucid dream experiences, their ability to control lucid dreams, and their training in lucid dream induction techniques. In addition to the monthly lucid dreaming frequency, the number of lucid dreams in the previous six-month period was assessed, which is a summative measure (i.e., an overall measurement taken after a period of time has passed) of lucid dreaming frequency as opposed to a formative approach (i.e., a measurement at shorter time intervals for each week or month). All items were presented either as an open text field or as a Likert-type format. The following single items were used: success of the lucid dream induction: “If you decide to have a lucid dream on a given night, how likely will you succeed?” (0 = *very unlikely*; 4 = *very likely*); wake-initiated lucid dream occurrences (0 = *never*; 4 = *always*); and how often one experiences a detached observer stance in the lucid dream (0 = *never*; 4 = *always*). To verify participants’ understanding of the lucid dream state, they were required to provide a brief report of one of their lucid dreams, detailing how they realized that they were dreaming. As all scales were adapted or created for this study, traditional reliability measures were not applicable. Nonetheless, a strong correlation was found between the frequency of lucid dreams per month and the number of lucid dreams in the previous six-month period (r_sp_ = 0.93, *p* < 0.0001). Participants who reported experiencing lucid dreams at least once per month were categorized as monthly frequent lucid dreamers (MFLDs), while those who reported experiencing lucid dreams at least once or more per week were classified as weekly frequent lucid dreamers (WFLDs), extending the standard classification convention [9,29,50,51].

Meditation Experience and Frequency. A revised version of the Meditation Experience Questionnaire [28] was utilized to assess the quantitative experience of meditation practices. To cover various meditation frameworks, three options of Buddhism (Theravadan, Tibetan, or Mixed) were extended to 18 different meditation frameworks: 9 religious/spiritual-oriented traditions (Theravadan, Tibetan, and Zen Buddhism; Daoism; Yoga; Sufism; Judaism; Christianity; and Shamanism) and 9 secular-oriented frameworks were included (app-guided, online-based, Vipassana, self-guided, Yoga, Thai Chi/Qigong, MBSR-based, non-dual meditation, and Transcendental Meditation), plus the option to specify an individual framework and tradition. Two items assessed previous meditation experience (*yes*/*no*), and meditation frequency. Meditation frequency was measured with a 16-point scale (0 = *never*, 15 = *more than 1 meditation per day*). Class means transformed the ordinal scale into metric units per week (0 → 0, 1 → 0.0185, 2 → 0.0192, 3 → 0.0385, 4 → 0.0769, 5 → 0.1346, 6 → 0.1923, 7 → 0.25, 8 → 0.50, 9 → 0.75, 10 → 1, 11 → 2, 12 → 3.5, 13 → 5.5, 14 → 7, 15 → 9). If participants meditated at least once per week, they reported how many different techniques they used in their meditation practice on a regular basis. In addition to questions about the quantitative meditation routine, they reported on the styles practiced while meditating, as well as meditation training and retreat experience. For each regularly practiced meditation framework, respondents provided an estimate of the length of an average meditation session in minutes, the frequency per day, the number of days per week, and the number of years of practice within the meditation framework. Based on this account, the total number of hours spent in meditation for each framework per week was calculated: $\frac{Minutes\;per\;Session\;*\;Freqency\;per\;Day\;*\;Days\;with\;Practice}{60}$. Moreover, the participants indicated the percentage of the total time they dedicated to each meditation framework in which a specific quality was facilitated: “Please indicate what percentage of your average meditation time you spend on a specific meditation technique”. Respondents were presented with six options: open monitoring (OM) meditation, focused attention (FA) meditation, loving-kindness/emotionally toned (LK) meditation, meditation to recognize the nature of the mind, non-dual meditation, and one option for an individually specified technique.

Meta-Awareness. The Meta-Awareness subscale of the Multidimensional Awareness Scale (MAS) captured the cognitive ability to recognize one’s current mental state based on self-assessment [52]. The item “I am aware of my thoughts and feelings as I experience them” reveals the direct aim of measuring the trait aspects of the cognitive process, as the instruction asked participants to indicate the extent to which the given statements represent the typical experience of their thoughts or feelings. The MAS-MA subscale consists of 7 items selected from the original 25-item MAS scale. The items were rated on a 7-point Likert scale (1 = *strongly disagree*, 7 = *strongly agree*). Reliability was found to be good, *ω_t_* = 0.89.

Trait Mindfulness. The measurement of trait mindfulness was performed using multiple instruments: the Toronto-Mindfulness Scale (TMS) [53,54], a short version of the Five Facet Mindfulness Questionnaire (FFMQ) [55,56], and the Relaxation, Meditation, and Mindfulness Experience Questionnaire (RMMtm) [36,57]. 

The TMS is a 13-item mindfulness inventory constructed as a 5-point Likert-scale (0 = *not at all*; 4 = *very much*). The original scale measured state mindfulness and the survey incorporated the trait version of the TMS, asking for experiences during the previous 7 days [53]. The scale was derived from a two-component definition of mindfulness: Curiosity and Decentering [1]. Curiosity includes 6 items and refers to the openness to explore one’s internal states as they occur. Decentering, measured by 7 items, refers to the ability to maintain a stance of detachment from one’s thoughts and emotions, with the capacity not to be carried away by thoughts and emotions [54]. The scales were found to be reliable, with *ω_t_* = 0.91 for Curiosity and *ω_t_* = 0.85 for Decentering.

Drawing from various definitions of mindfulness, a psychometrically validated trait mindfulness measure, the FFMQ, was developed. It combines five mindfulness instruments, and factor analysis on a large sample revealed five factors [58]. Its condensed form has 20 items rated on a 5-point Likert scale (1 = *never or rarely true*, 5 = *very often or always true*) [56]. The brief version measures five distinct but related components of mindfulness, with 4 items for each scale: (1) Observing: the ability to notice and attend to internal and external experiences, *ω_t_* = 0.76; (2) Describing: the capacity to articulate one’s experiences in words, *ω_t_* = 0.77; (3) Acting with awareness: as opposed to “being on autopilot”, the degree of presence and awareness while engaged in activities, *ω_t_* = 0.88; (4) Nonjudgment of inner experience: the ability to refrain from judging inner experiences as neither good nor bad, *ω_t_* = 0.88; (5) Nonreactivity to inner experience: the attitude of allowing thoughts and feelings to arise and pass without being caught up in or swept away by them, allowing these experiences to exist without interference, *ω_t_* = 0.77. The psychometric properties of the short version of the FFMQ by Tran et al. [56] demonstrated moderate to strong reliability.

The Relaxation, Meditation, and Mindfulness Tracker (RMMtm) by Smith [59] comprises a self-reported broad-spectrum inventory for assessing dispositional mindfulness. The 7-point Likert scale with 32 items captures trait mindfulness (1 = *never*, 2 = *not this month, but once or twice this year*; 3 = *about once this month*; 4 *= about once a week*; 5 = *about 2 or 3 times a week*; 6 *= about every day*; 7 = *several times daily*). The RMMtm includes various experiences associated with the practice of mindfulness meditation techniques that manifest as characteristics over time. The instrument was derived from third-wave mindfulness theory to capture all essential phenomenological states of mindfulness, represented on a continuum with varying levels of mindfulness [60]: (1) Mindful Relaxation, which assesses initial mindful relaxation experiences; (2) Mindful Quiet Focus, which captures the stillness and observational nature of meditation; (3) Mindful Engagement, which measures the ability to remain present and compassionate during activities; and (4) Mindful Transcendence, the deepest level observed in long-term practitioners in spiritual or non-secular contexts [61]. Due to expected variations in levels of mindfulness among long-term and frequent meditators compared to other groups, separate component analysis for each population sample is recommended [36,57]. A principal component analysis identified 2 distinct factors. Items 1–24 loaded on the first factor; this dimension is interpreted as Mindful Relaxation and Focus, *ω_t_* = 0.97. The first dimension includes aspects of physical relaxation but also cognitive and emotional aspects of mindfulness: “I was living in the present moment, not past or future concerns” or “I felt selfless/caring/compassion”. Items 25–32 loaded on the second factor, compromising Mindful Transcendence, *ω_t_* = 0.93. Self-transcendence, in general, is defined as the capacity to expand self-boundaries [62]. The subscale refers to transpersonal self-transcendence: “I had a sense of what is timeless, boundless, infinite”. Due to the sample-specific component analysis, reliability measures were not comparable to previous research. However, reliability analysis indicated satisfying internal consistencies.

### 2.4. Statistical Analysis

Statistical analyses were performed using R, version 4.3.1, and SPSS, version 29. For the primary data analysis, only the general mixed sample was utilized, following manual validation of each participant’s lucid dream report. The student populations were not included in the data analysis for the research questions addressed in this study, since there was an insufficient number of participants per group. Data management was based on the functions of the tidyverse package in R [63]. A total of 270 participants out of 291 were considered eligible for data analysis. The more liberal Shapiro–Wilk tests revealed significant deviations from normal distribution for most of the measures, e.g., lucid dreaming frequency (Shapiro–Wilk’s W = 0.590, *p* < 10^−14^), dream recall frequency (Shapiro–Wilk’s W = 0.855, *p* < 10^−15^), lucid dream induction frequency (Shapiro–Wilk’s W = 0.540, *p* < 10^−22^), and meditation frequency (Shapiro–Wilk’s W = 0.813, *p* < 10^−16^). Therefore, non-parametric independent two-sample permutation tests were utilized for assessing overall group differences. For each group, a set of multiple tests with adjusted *p*-values based on Benjamini and Hochberg and the False Discovery Rate (FDR) correction were reported [64]. 

Monte Carlo permutation tests with R = 10,000 permutations were implemented for group comparisons [65]. For most group comparisons, the Monte Carlo permutation test of the mean was calculated. All variables besides the ordinal scaled WILD frequency and the ordinal variable related to the detached observer stance were continuous variables. When comparing the total hours of meditation per week and the largest number of years for a meditation practice, a Monte Carlo permutation test of the median was used. In addition to that, Spearman’s correlation was preferred over the Pearson correlation due to the influence of the largest values in the weekly hours of meditation for each meditation practice style.

Prior to this, an exploratory principal component analysis found the sample-specific RMMtm scales [36]. Bartlett’s test of sphericity was significant (*X*^2^ = 8357.33, df = 496, *p* < 0.001), indicating that the variables were sufficiently intercorrelated to proceed with principal component analysis (PCA). A PCA with Kaiser normalization and an Oblimin rotation method yielded a two-factor solution in the general population with tools from the psych package [66]. The Kaiser–Meyer–Olkin Measure was 0.953, which indicates good sampling adequacy. Factor extraction was based on the scree plot, indicating two factors, while parallel analysis yielded two factors and Kaiser–Gutman criteria indicated four factors. The two-factor solution accounted for 62% of the cumulative variance, compared to 67% for the three-factor and 70% for the four-factor solution. The tools from the MBESS package calculated omega total (*ω*_t_) estimates instead of the psych tools for a more conservative reliability measure [43,47].

Multiple regression analysis for the monthly lucid dreaming frequency, predicted by the RMMtm Mindful Transcendence subscale, the TMS Decentering and Curiosity subscale, the MAS-MA subscale, weekly meditation frequency, and age, was implemented with the Boot and LessR package in R (i.e., Appendix A) [67,68]. Due to violations of heteroscedasticity and normal distribution, examined by means of the visual plotting of the predicted values against the standardized residuals, the coefficients were tested based on 10,000 bootstrapped bias-corrected confidence intervals. Assumptions were investigated via the LessR package: no VIF values exceeded 5, and none of Cook’s distance indexes was larger than 1, with the highest VIF being 3.634 and the largest Cook’s distance index being 0.12. 

The exploratory analysis of the relationship between lucid dreaming, meditation, and meta-awareness was performed with model 4 of the PROCESS macro for R [69]. In this model, meta-awareness was used as a mediator, meditation frequency as the independent variable, and the number of lucid dreams in the previous six months was used as the outcome variable. The model controlled for covariates including lucid dream induction frequency, dream recall frequency, and age. It must be stated that for this analysis, 25 participants who did not have prior experience with lucid dreams were excluded, due to missing data. Bias-corrected accelerated (BCa) confidence intervals for the coefficients were bootstrapped with R = 10,000 replicates, as was the confidence interval for the indirect effect. Standard errors were computed using heteroscedasticity-consistent estimates due to the heteroscedasticity of the residuals and the nonnormality of the dependent variable (Shapiro–Wilk’s W = 0.381, *p* < 10^−22^). The model parameters were standardized, and the random seed was fixed to 9999 to ensure the reproducibility of the results.

## 3. Results

In the general sample spanning all groups, participants reported an average of 5.92 ± 3.03 [M ± SD, *n* = 270] dreams recalled weekly and 4.26 ± 7.65 [M ± SD, *n* = 270] remembered lucid dreams per month. A crucial number of participants had at least one lucid dream (90.74%), induced 3.74 ± 7.72 [M ± SD, *n* = 245] lucid dreams each month, and indicated an average of 27.80 lucid dreams over the previous half year (SD = 71.90, Range = 0–720). A total of 143 participants (52.96%) had at least one lucid dream per month, with an average of 7.80 ± 9.16 [M ± SD] lucid dreams per month, defined as monthly frequent lucid dreamers, whereas 73 experienced at least one lucid dream per week (27.04%), with an average of 13.70 ± 9.67 [M ± SD] monthly lucid dreams, and thus were classified as WFLDs; see Table 1. 

Across the whole general sample, bivariate correlations between dream variables yielded a significant positive correlation between weekly dream recall and monthly lucid dreaming frequency (r_sp_ = 0.44, *p* < 0.001) as well as lucid dreams in the previous six months (r_sp_ = 0.36, *p* < 0.001). In addition, monthly lucid dream induction frequency correlated with the monthly lucid dreaming frequency (r_sp_ = 0.46, *p* < 0.001) and the estimated lucid dreams in the previous six months (r_sp_ = 0.47, *p* < 0.001). Furthermore, self-reported meta-awareness correlated positively with monthly lucid dreams (r_sp_ = 0.23, *p* < 0.001), while less strongly with the summative measure of lucid dreams in the previous six months (r_sp_ = 0.14, *p* = 0.012). Weekly meditation frequency and age were not significantly associated with either the monthly lucid dreaming frequency or the total number of lucid dreams in the most recent six-month period (all *p* > 0.05, Table 2). However, weekly meditation frequency was significantly associated with age (r_sp_ = 0.38, *p* < 0.001), meta-awareness (r_sp_ = 0.25, *p* < 0.001), and the frequency of wake-initiated lucid dreams (r_sp_ = 0.15, *p* < 0.001).

Out of 270 respondents, 243 (90.01%) meditated at least once, of which 91 (33.70%) meditated infrequently, with 0.22 ± 0.26 [M ± SD] weekly meditations. A total of 152 respondents (56.30%) reported meditating at least once a week; the average weekly meditation frequency was 5.58 (SD = 2.72, Range = 1–9). Among the frequent meditators, 35 practiced up to multiple times daily (12.96%), termed DFMs. A total of 13 WFMs and 4 DFMs violated the instructions of the survey, leading to an unrealistic meditation practice time (i.e., 0 minutes of meditation or more than 50 hours per week). Hence, they were excluded from specific analyses. A total of 135 weekly meditators stated a median experience of 7.1 years (SD = 13.7, Range = 0.5–55) in their most consistently practiced meditation framework and a median of 4.67 practiced meditation hours weekly (SD = 7.3, Range = 0.133–45). When looking at the specific meditation styles and techniques, the average total hours per week were divided into six different meditation techniques. The highest practiced minutes averaged over frameworks and traditions was non-dual meditation with 47.44 ± 107.27 [M ± SD], and then OM meditation with 47.01 ± 96.22 [M ± SD], closely followed by FA meditation with 42.07 ± 80.14 [M ± SD], meditation to recognize the true nature of the mind with 17.82 ± 72.32 [M ± SD], LK meditation to cultivate emotional capacities with 20.5 ± 50.3 [M ± SD], and other contemplative techniques with 8.34 ± 25.75 [M ± SD].

Meditation frameworks diverged, falling either within traditions with spiritual and religious backgrounds or within secular frameworks. On average, participants engaged in 2.65 ± 1.96 [M ± SD, *n* = 141, Range = 1–11] traditions or frameworks for their meditation practices. Most participants meditated in a self-guided (69) and an online-based (40) setting, followed by meditation included in secular Yoga practices (29), app-guided meditation (27), Vipassana (24), Transcendental Meditation (15), non-dual meditation (15), meditation included in secular Thai Chi/ Qigong practices (12) and mindfulness-based stress reduction (MBSR) meditation (8). Within the non-secular traditions, most practitioners were rooted in Tibetan Buddhism (34), Zen Buddhism (29), and religious or spiritual Yoga practices (15), as well as Theravadan Buddhism (14), followed by Shamanistic (9) and Christian (5) meditation techniques, concluding with meditation rooted in Daoism-based (2) and Sufi traditions (2). Notably, Judaistic meditation practices were not represented. Participants also had the flexibility to specify an additional tradition (10) or secular approach (14).

### 3.1. Lucid Dreaming in Meditators

All frequent meditators together (collapsing DFMs and WFMs) had an insignificantly higher monthly lucid dreaming frequency compared to non-frequent meditators. In addition, the numbers of lucid dreams in the previous six-month period revealed insignificantly higher numbers of lucid dreams in frequent meditators compared to non-frequent meditators (see Table 3, right panel), whereas the dream recall frequency was significantly higher in frequent meditators than in non-frequent meditators. All frequent meditators had significantly more wake-initiated lucid dreams compared to the WILD occurrences in non-frequent meditators, while also being more often in the role of a detached observer in the lucid dream compared to non-frequent meditators. Having control in the lucid dream, as is possible in waking life, showed no significant difference between frequent meditators and the rest of the participants (*p* > 0.05; Table 3, right panel). 

Among the weekly meditators, 45 practitioners reported experiencing lucid dreams once or more per week. When correcting for the hours spent in meditation, 38 WFLD frequent meditators dedicated significantly more time to meditation each week (Mdn = 5.29, SD = 9.29) in comparison to their NWLD frequent meditating counterparts (Mdn = 3.67, SD = 6.32, *n* = 97) [*p* = 0.039; Monte Carlo permutation test]. The number of years of meditation experience for WFLD frequent meditators (Mdn = 7.00, *SD* = 13.84) did not differ significantly from NWLD frequent meditators (Mdn = 6.00, SD = 13.50) [*p* = 0.400; Monte Carlo permutation test].

When looking at the group with more intensive meditation practice, DFMs had significantly more lucid dreams per month compared to non-frequent meditators; see Table 3, left panel. In addition, there were significantly higher numbers of lucid dreams in the previous six-month period in DFMs compared to non-frequent meditators. Dream recall frequency differed significantly from non-frequent meditators. DFMs were less often controlling in the way that is possible during waking life compared to non-frequent meditators. Moreover, DFMs were significantly more often in the detached observer stance in a lucid dream than non-frequent meditators. Furthermore, DFMs experienced significantly more compared to non-frequent meditators; see Table 3, left panel. 

After excluding the non-valid cases, DFMs reached significance with a median of 9.33 ± 7.52 [Mdn ± SD, *n* = 31] hours of weekly meditation compared to 3.50 ± 6.80 [Mdn ± SD, *n* = 104] practiced hours in WFMs [*p* < 0.0001; Monte Carlo permutation test]. There was no significant difference in meditation experience in years (*p* > 0.05). A total of 12 DFMs who were classified as WFLDs engaged in 3.58 ± 2.35 [M ± SD] meditation frameworks or traditions: self-guided meditation (5), Transcendental Meditation/TM (5), meditation from Shamanistic traditions (4), Vipassana meditation (4), Tibetan (4), Zen (4), and Theravadan (2) Buddhism, non-dual meditation (3), and meditation in secular Thai Chi/ Qigong practices (4). There was no significant difference in the total hours of meditation per week between WFLDs and NWLDs in daily frequent meditators (*p* > 0.05). DFMs who lucid-dream weekly had significantly more meditation experience, with a median of 25 ± 16.85 years [Mdn ± SD, *n* = 11], compared to non-weekly lucid dreaming DFMs, with a median of 5 ± 14.66 years (Mdn ± SD, *n* = 19) [*p* = 0.029; Monte Carlo permutation test]. 

### 3.2. Lucid Dreaming and Meditation Practices

The assessment of the varied qualities cultivated during meditation, based on specific meditative practices, was accomplished by distributing the total weekly practice hours across the percentage of the techniques each participant practiced, and then averaging the practice time for these qualities/techniques across all traditions for each participant. As hypothesized, the open monitoring meditation practice exhibited a significant positive bivariate correlation with the number of lucid dreams in the previous six-month period (r_sp_ = 0.16, *p* = 0.037). Hence, more weekly practiced hours of OM meditation were associated with more lucid dreams per month. Other techniques did not show any significant relationship: FA meditation (r_sp_ = 0.08, *p* > 0.05), meditation related to the nature of the mind (r_sp_ = 0.08, *p* > 0.05), nondual meditation (r_sp_ = 0.06, *p* > 0.05), and LK meditation (r_sp_ = 0.03, *p* > 0.05) showed insignificant bivariate correlations with the number of lucid dreams during the previous six months. Furthermore, when looking at the association between meditation practices and monthly lucid dreaming frequency, there was no significant bivariate association (all *p* > 0.05).

### 3.3. Lucid Dreaming and Mindfulness Instruments

Weekly frequent lucid dreamers descriptively scored more highly in all mindfulness measurements, except for the Describing facet of the FFMQ. If collapsing participants across all groups (DFMs, WFMs, and non-frequent meditators): WFLD (M = 3.55, SD = 0.79) exceeded NWLD (M = 3.23, SD = 0.77) in the Nonreactivity subscale of the FFMQ [*p* = 0.018; Monte Carlo permutation test with FDR correction]. Furthermore, the sample-specific Mindful Transcendence subscale was also higher in WFLDs (M = 3.59, SD = 1.69) compared to NWLDs (M = 3.05, SD = 1.56) [*p* = 0.027; Monte Carlo permutation test with FDR correction]. Within non-frequent meditators, no differences reached significance after controlling for multiple testing when comparing WFLDs and NWLDs (all *p* > 0.05; Table 4, right panel). 

Within the frequent meditators, WFLDs scored descriptively higher in all mindfulness measurements but none of the differences reached statistical significance after controlling for multiple testing (all *p* > 0.05; Table 4, middle panel). In contrast, within the DFMs, WFLDs surpassed NWLDs in the FMMQ mindfulness aspects of Nonreactivity, Describing, and Observing (Table 4, left panel).

### 3.4. Individual Differences in Meta-Awareness

Meta-awareness differed significantly between meditators: DFMs scored highest at 6.06 ± 0.68 [M ± SD] compared to WFMs at 5.7 ± 0.91 [M ± SD] (*p* = 0.0224) and non-frequent meditators at 5.34 ± 1.04 [M ± SD] (*p* = 0.0007). In addition, meta-awareness scores between WFMs and non-frequent meditators reached significance [*p* = 0.0035; Monte Carlo permutation test with FDR correction]. Separately, across the general sample for the lucid dreaming groups, meta-awareness was highest in WFLDs, scoring 5.90 ± 0.98 [M ± SD], compared to monthly frequent lucid dreamers, who scored 5.56 ± 0.90 [M ± SD] (*p* = 0.0247), and non-frequent lucid dreamers, who scored 5.44 ± 0.98 [M ± SD] (*p* = 0.0032) [Monte Carlo permutation tests with FDR correction]. However, monthly frequent lucid dreamers could not reach significantly higher scores in meta-awareness compared to non-frequent lucid dreamers [*p* = 0.2256; Monte Carlo permutation test with FDR correction]. Within the non-frequent meditators, WFLDs also showed higher values with 5.82 ± 0.91 [M ± SD] in meta-awareness compared to NWLDs with 5.20 ± 1.03 [M ± SD] [*p* = 0.0047; Monte Carlo permutation test with FDR correction]. In contrast, WFLDs within the weekly frequent meditators could not reach significantly higher values, 5.73 ± 1.13 [M ± SD], than NWLDs who meditate once a week, 5.71 ± 0.83 [M ± SD] (*p* = 0.4346). However, DFMs who are weekly frequent lucid dreamers scored significantly higher, 6.44 ± 0.48 [M ± SD], compared to non-weekly lucid DFMs, 5.83 ± 0.69 [M ± SD] (*p* = 0.0079) [Monte Carlo permutation test with FDR correction].

### 3.5. Meditation Frequency, Mindfulness, and Meta-Awareness for Lucid Dreaming

Mindful Transcendence, measured using the RMMtm, and the two dimensions, Decentering and Curiosity, of the TMS, as well as the MAS subscale measuring meta-awareness, in addition to weekly meditation frequency, were utilized to predict monthly lucid dreaming frequency. Therefore, a multiple linear regression model with BCa bootstrapped confidence intervals was performed, given the violation of the assumptions of normality and homoscedasticity of the residual. All predictors accounted for R^2^ = 0.098, F (6, 263) = 4.780, *p* < 0.001. Transcendence [b = 1.228, BCa 95% CI (0.423, 1.219)], in addition to the MAS-MA subscale [b = 1.420, BCa 95% CI (0.633, 1.405)], had significant predictive regression coefficients for the monthly frequency of lucid dreams (Table 5). Age was statistically controlled for and did not have a significant coefficient. Therefore, higher scores on the Transcendence subscale, as well as higher scores on the MAS-MA meta-awareness subscale, are associated with a higher frequency of lucid dreams per month.

Two regression models and one mediation analysis were conducted to further elucidate the relevance of meta-awareness for the association between meditation frequency and lucid dreaming frequency. Specifically, the relationship between weekly meditation frequency and the number of lucid dreams in the previous six-month period (Lucid Dreams 6M) was tested with meta-awareness as the mediator. Weekly dream recall frequency, age, and monthly lucid dream induction frequency were included as covariates. Considering the violations of normality, homoscedasticity, and concerns with leverage and outliers, the coefficients’ confidence intervals from both regression models, as well as the test of the indirect path, were bootstrapped with R = 10,000 runs, and a heteroscedasticity-consistent standard error was used.

The first model, with meta-awareness as the outcome variable, explained a total of R^2^ = 0.124 variance, F (4, 240) = 9.836, *p* < 0.0001. Age, dream recall, and meditation frequency significantly predicted meta-awareness (Table 6). 

More frequent meditation was associated with higher meta-awareness (b = 0.048, BCa 95% CI [0.009, 0.086]). Dream recall frequency (b = 0.047, BCa 95% CI [0.004, 0.091]) and age (b = 0.013, BCa 95% CI [0.004, 0.091]) were both significantly positively correlated with meta-awareness. Predicting lucid dreaming frequency (Lucid Dreams 6M) explained R^2^ = 0.305 variance, F (5, 239) = 4.778, *p* < 0.0001. The mediator analysis of the coefficient product of the indirect path, the coefficient of predicting meta-awareness based on meditation frequency, and Lucid Dreams 6M based on meta-awareness resulted in a significant positive indirect effect (b = 0.303, BCa 95% CI [0.013, 1.009]; see Figure 1. The direct effect of meditation frequency on Lucid Dreams 6M revealed an insignificant positive effect (b = 0.703, 95% CI [−2.030, 3.435]; *p* = 0.613). Two covariates, higher lucid dream induction frequency (b = 4.913, BCa 95% CI [2.357, 8.296]) and younger age (b = −0.357, BCa 95% CI [−0.857, −0.037]), were significantly associated with higher numbers of lucid dreams in the previous six months.

## 4. Discussion

The present study explored the link between lucid dreaming, dispositional mindfulness, and meditation practices, emphasizing the role of meta-awareness. The results from this study indicate that individuals who meditate more than once per day have more lucid dreams compared to infrequent meditators. Therefore, the hypothesis that frequent meditation is associated with more lucid dreams was supported, adding to the already existing body of work [28,33]. The results also partially confirmed the hypothesis that open monitoring meditation was positively associated with increased lucid dreaming. In addition, meta-awareness was found to be highest in daily frequent meditators and was also elevated in weekly lucid dreamers without meditation experience. Together, these findings indicate a link between lucid dreaming, meta-awareness, and OM meditation. As OM meditation is known to enhance sustained open awareness, this supports the idea that non-propositional and sustained meta-awareness could be a key capacity for lucid dreaming [7]. Exploratory analyses highlighted the mediating role of meta-awareness in the relationship between weekly meditation frequency and lucid dream occurrences during the previous six months. Furthermore, the evidence that aspects of trait mindfulness are associated with frequent lucid dreaming points towards the continuity of mindful awareness from waking consciousness to sleeping consciousness [70,71]. In addition to meta-awareness, transcendence also emerged as a positive predictor of monthly lucid dreaming frequency across all groups, replicating previous research [36]. 

Several conceptual links have been made between lucid dreaming and meditation practices, postulated to be influenced by regulating attention and meta-awareness [19,28,33]. A meta-analysis looked at meditation practices across various traditions and backgrounds and extracted the various effects of different meditation styles and their influence on cognitive capacities [72]. Several meditation practices influence the dynamics and direction of attention: in FA meditation, attention is directed towards one object, e.g., focusing on the breath and keeping the concentration on the same object up to the whole session of meditation [17]. This process involves meta-awareness to recognize the wandering mind. Once the inner focus is distracted, e.g., by thoughts about the future, meta-awareness can detect the distraction and thus return to the initial object of focus [8,18]. In some practice styles, OM meditation is postulated to evolve from FA meditation [17]. As individuals progress in their meditation practice, they cultivate monitoring skills that become the crucial point to the practice of OM meditation. The practitioner attempts to remain in a state of pure observation, vigilantly attending and monitoring each moment-to-moment event in awareness without anchoring the focus on any particular object [17].

Existing studies have not examined the specific meditative practices that might lead to more frequent lucid dreams. The purpose of this study was to study the link between specific meditation practices and lucid dreams by investigating the time spent on five different meditation styles per week. The results partially confirmed the main hypothesis that FA meditation and OM meditation would correlate with an increased number of monthly lucid dreams, with a positive association observed only between the duration of averaged weekly OM practice and the number of lucid dreams over the previous six months. 

However, it should be noted that there was a discrepancy between the formative and summative measurement of lucid dreaming frequency. Specifically, in contrast to the lucid dreams in the previous 6 months, there was no significant association between the average time spent in OM meditation and average monthly lucid dreaming frequency among meditators. One reason for this discrepancy may be that the 6-month measure is a more accurate assessment of lucid dreaming frequency as it is more precise (it requires that participants report a specific number of lucid dreams) and over a proximal, well-defined time interval (the previous 6 months). Another possibility is that meditation experience and practice could influence lucid dreaming, but possibly only in individuals who have already had lucid dreams. This is consistent with research where meditation has been successfully integrated as a complementary technique in combination with cognitive and substance-enhanced lucid dream induction methods [73]. 

Only a handful of studies have examined the relationship between meditation experience and lucid dreaming frequency [26,28,29,31,33,34,35,36,74]. Baird and colleagues [28] observed a higher frequency of spontaneous lucid dreams per month among long-term meditators compared to those with less meditation experience. The current study supports these findings, with daily frequent meditators reporting a greater number of lucid dreams per month than infrequent meditators. Consequently, this study supports the notion that regular and intensive meditation is associated with an increased incidence of lucid dreams [28]. In particular, it was found that meditating several times a day was associated with a weekly occurrence of lucid dreams, a finding that is consistent with results observed in practitioners of Transcendental Meditation [29]. 

On a cognitive, psychological, and neuropsychological level, researchers have suggested that the link between lucid dreaming and meditation could be an increase in self-reflectiveness and meta-awareness [28,31,33,75]. Given that no earlier research has examined individual differences in self-reported meta-awareness within and between lucid dreamers and meditators, this study analyzed individual differences in meta-awareness using the MAS-MA scale. Daily frequent meditators reported the highest self-rated meta-awareness on this scale within the general sample. Furthermore, weekly lucid dreamers showed higher meta-awareness scores compared to both monthly and infrequent lucid dreamers. Monthly frequent lucid dreamers showed only a marginal increase in meta-awareness compared to infrequent lucid dreamers across the sample. Notably, even among non-meditators, weekly lucid dreamers expressed higher meta-awareness than non-weekly lucid dreamers. Only within the DFM subgroup was weekly lucid dreaming associated with higher meta-awareness compared to non-weekly lucid dreamers. These results implicate an effect of meditation on meta-awareness, as well as an effect of meta-awareness on lucid dreaming, pointing towards a mediating effect of meta-awareness on lucid dreaming frequency. 

Indeed, the results of an exploratory mediation analysis supported the hypothesis that meta-awareness fully mediates the positive association between meditation frequency per week and the number of lucid dreams in the previous six-month period. However, it has to be noted that a mediation model with this study design cannot imply a causal relationship since the variables were assessed simultaneously, and there might be other variables and cognitive mechanisms involved that were not captured. Nonetheless, the findings are consistent with the hypothesis that higher meta-awareness is associated with more lucid dreams [7,19].

Consistent with previous findings by Stumbrys et al. [33] and Baird et al. [28], the results suggest that lucid dreaming is associated with specific aspects of mindfulness that differ depending on whether an individual has experience with meditation. WFLDs reported higher scores on the Non-reactivity subscale of the Five Facet Mindfulness Questionnaire (FFMQ) than did non-weekly lucid dreamers. Nonreactivity involves the ability to experience thoughts and feelings without becoming caught up in them [1]. Higher scores in this area for WFLDs may indicate a more developed ability to observe experiences without immediate reaction, which may be advantageous for recognizing the dream state and becoming lucid, without being caught up in the event and thereby losing lucidity. Further analysis showed that WFLDs also had significantly higher scores on the RMMtm Mindful Transcendence subscale compared to NWLDs. This facet reflects a heightened awareness of the present experience and a more transcendent perspective that may facilitate the detachment necessary for dream lucidity. Mindful Transcendence measured in this study is comparable to the transcendence subscale found in the sample of students in the study by Geise and Smith [36]. We replicated that feeling more self-transcendent, like “I felt connected. I felt at one with everything and humanity. I felt in harmony with the world. I felt a sense of belonging. A part of something larger”, positively predicted lucid dreaming frequency per month [36]. In contrast, there were no significant differences in mindfulness facets found between WFLDs and NWLDs within the group of non-frequent meditators and all frequent meditators, DFMs and WFMs together, after adjustment for multiple comparisons. This is in contrast with the study of long-term meditators, where non-frequent meditators scored higher on the Describing scale of the FFMQ [28]. 

There has been no other study to date that investigated aspects of lucid dream experiences in meditators. Here, we found that daily frequent meditators reported having more wake-initiated lucid dreams, were more often in the observing stance in a lucid dream, and were less often actively exerting ordinary forms of control of the dream compared to infrequent meditators. Overall, these results support the notion that consistent, intensive meditation practice may enhance one’s ability to maintain a nonreactive, observing stance, and, by extension, enhance the state shift from ordinary to lucid dreams. The results indicate that the frequency and depth of mindfulness practices are linked to the experience of lucid dreaming and underscore the importance of considering individual differences in meditation practice and experience for lucid dreaming. 

## 5. Limitations and Future Directions

It is crucial to highlight that the general mixed sample included a substantial proportion of frequent meditators, 56% of respondents, as well as an extremely high number of monthly frequent (53%) and weekly frequent (27%) lucid dreamers. This distribution does not reflect the general population, a discrepancy that is highlighted when compared to student populations or representative cohort studies [49,51]. Meta-analytic research suggests a 23% prevalence of monthly lucid dreams and a 55% likelihood of having at least one lucid dream [76]. The overrepresentation in this study is likely due to self-selection bias, disproportionately including individuals interested in lucid dreaming, dreaming, and meditation [77]. 

The introduction of a 16-point Likert scale marked a major advance in the assessment of lucid dreaming frequency, providing more variability but making direct comparisons with responses from previous studies more difficult. Many studies operationalized lucid dreaming frequency based on the scale developed by Schredl and Erlacher [78] as a formative measure (giving an estimate of how many lucid dreams one has per month), but few studies also incorporated a summative approach to measuring lucid dreams [31]. The correlation analysis related to the hours spent in OM meditation showed a discrepancy between the two measures of lucid dreaming frequency, although the association between the two frequency measures was high. The discrepancy needs to be considered when interpreting the relationship between lucid dreams and meditation practices.

The cross-sectional nature of the study with a single data collection point limits conclusions to correlational and associative implications. Biases such as social desirability and performance bias, particularly prevalent in a convenience sample likely to be interested in meditation and (lucid) dreaming, may bias self-reported measures of meta-awareness and mindfulness [79]. This potential response bias and selective sample underscore the need for caution when interpreting the results. Recall bias presents a major challenge in retrospective assessments, potentially distorting the accuracy of participant reports of their experiences of lucid dreaming and meditation practices [80]. This bias can be particularly problematic when relying on individuals to estimate the frequency of events over extended periods of time. Along this line, dream recall is usually underestimated in retrospective measurements compared to daily log-books [81]. Ecological Momentary Assessment (EMA) could mitigate this problem by collecting data in real time, thereby providing more accurate and immediate reports of experiences as they occur [82]. Implementing EMA in future studies would not only increase the reliability of self-reported data but also provide greater insight into the dynamic interplay between mindfulness practices and lucid dreaming. Future investigations should include a wide scale for lucid dreaming frequency, encompassing the higher end of variability in lucid dreamers. Also, it could be fruitful to compare different meditation practices within different frameworks and traditions (e.g., app-based meditation vs. Transcendental Meditation).

## 6. Conclusions

The findings of this study validate the association between frequent meditation, specifically open monitoring (OM) meditation, and increased lucid dreaming frequency, and support a role of meta-awareness in enhancing lucid dream experiences. These results suggest that OM meditation enhances sustained meta-awareness, which is essential for recognizing and maintaining lucidity in dreams. It would be worthwhile in future research to test this hypothesis through a random-assignment meditation intervention pre-post design study. In addition, the experiences of expert meditators and those practicing dream Yoga should be explored to further understand consciousness in different sleep states. Longitudinal studies and intensive retreats may prove valuable in assessing the effects of meditation on lucid dreaming. Complementary methods, such as sleep diaries and EMA, could allow for the detailed tracking of lucid dreams and meditation practices. Investigating neurophysiological changes of expert meditators during sleep might also shed light on the neural underpinnings of meditation-related changes in consciousness.

## Figures and Tables

**Figure 1 brainsci-14-00496-f001:**
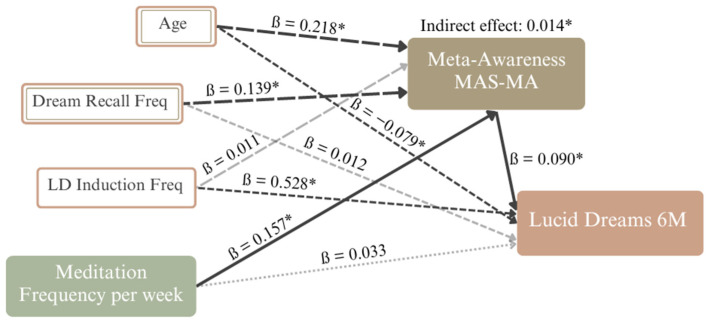
Complete mediation model for the relationship between meditation frequency and the total number of lucid dreams in the previous six-month period with age, weekly dream recall, and monthly lucid dream induction frequencies as covariates. Note: *n* = 245. Bootstrapped BCa with R = 10,000. The model includes only standardized coefficients and effects; * = significant effect; grey dotted line = the insignificant direct effect; large dashed grey line = the insignificant effect of the covariate on the mediator; small grey dashed line = the effect of the covariate on the dependent variable; small dashed black lines = the significant effects of the covariates on the dependent variable; large dashed black lines = the significant effects of the covariates on the mediator; solid black lines = the indirect path.

**Table 1 brainsci-14-00496-t001:** Number of participants in groups based on lucid dreaming and meditation frequency.

	Frequent Meditators		Non-Frequent Meditators		
	DFMs	WFMs	Infrequent Meditators	Never Have Meditated	
NWLDs	22	85	72	18	
	107		90		*n* = 197
WFLDs	13	32	19	9	
	45		28		*n* = 73
	35	117	91	27	
	*n* = 152		*n* = 118		*n* = 270

Note: WFLDs = weekly lucid dreamers; NWLDs = non-weekly lucid dreamers; DFMs = daily frequent meditators; WFMs = weekly frequent meditators.

**Table 2 brainsci-14-00496-t002:** Descriptive statistics and bivariate intercorrelations of important variables.

	M (SD)	DRF	M LDF	6M LDF	LDIF	Induce S.	WILD	META	Medi
Weekly Dream Recall	5.92 (3.03)								
Monthly LDF	4.26 (7.65)	0.47 **							
6-Month LDF ^a^	27.79 (71.89)	0.35 **	0.93 **						
LD Induction F ^a^	3.74 (7.72)	0.45 **	0.47 **	0.47 **					
Induction Success ^a^	1.5 (1.12)	0.32 **	0.66 **	0.62 **	0.42 **				
WILD Freq ^a^	1.31 (1.15)	0.08	0.53 **	0.51 **	0.28 **	0.57 **			
MAS-MA	5.60 (0.97)	0.19 *	0.23 **	0.15 **	0.09	0.28 **	0.22 **		
Weekly Meditation Freq	3.24 (3.36)	0.14 *	0.06	−0.02	0.06	0.09	0.15 **	0.25 **	
Age	37.74 (16.16)	−0.07	−0.01	−0.06	−0.07	0.02	0.13	0.25 **	0.38 **

Note. ** = *p* < 0.001; * = *p* < 0.05; *n* = 270; ^a^ = 25 participants excluded; Correlations based on Spearman’s correlation coefficient; Monthly LDF (M LDF) = Monthly lucid dreaming frequency; 6-Month LDF ^a^ (6M LDF) = Total number of lucid dreams during the most recent six-month period; LD Induction F ^a^ (Induce S.) = Monthly lucid dream induction frequency; Induction Success ^a^ (0 = *very unlikely*; 4 = *very likely*); WILD Freq ^a^ (WILD) = Frequency of wake-initiated lucid dreams (0 = *never*; 4 = *always*); DRF = Weekly dream recall frequency; MAS-MA (META) = Meta-Awareness subscale; Weekly Meditation Freq (Medi) = Weekly meditation frequency.

**Table 3 brainsci-14-00496-t003:** Group comparisons for lucid dream experience variables between different meditation groups.

	DFMs > Non-Frequent Medi.				Frequent Medi. > Non-Frequent Medi.			
	(*n* = 35)		(*n* = 118)		(*n* = 152)		(*n* = 118)	
	M (SD)	M (SD)	M Diff	*p* ^a^	M (SD)	M (SD)	M Diff	*p* ^a^
Frequency								
LDF 6M	56.7 (156.6)	20.4 (41.9)	33.1	0.03 *	33.3 (87.7)	20.4 (41.9)	12.84	0.11
LDF monthly	6.44 (9.1)	3.55 (7.0)	2.89	0.03 *	4.80 (8.1)	3.55 (7.0)	1.25	0.11
DRF weekly	7.04 (2.5)	5.40 (3.1)	1.65	0.01 *	6.33 (2.9)	5.40 (3.1)	0.92	0.02 *
WILD	1.64 (1.4)	1.14 (1.1)	0.50	0.03 *	1.44 (1.2)	1.14 (1.1)	0.29	0.04 *
Control								
Control PWL	2.03 (1.1)	2.57 (1.1)	−0.54	0.03 *	2.38 (1.2)	2.57 (1.1)	−0.19	0.11
Experience								
Detached observer	1.52 (1.0)	1.10 (1.0)	0.42	0.03 *	1.56 (1.0)	1.10 (1.1)	0.46	0.0006 *

Note: M Diff = Mean difference between observed mean and mean constructed under the null hypothesis with R = 10,000 permutations; *p*
^a^ = *p*-value corrected with false discovery error rate, * = *p* < 0.05; DFMs = daily frequent meditators; Frequent-Medi = Frequent meditators; LDF 6M = Number of lucid dreams during the previous six-month period, LDF monthly = Monthly lucid dreaming frequency, DRF weekly = Weekly dream recall frequency; WILD = Frequency of wake-initiated lucid dreams; Control PWL = Control aspects of the lucid dream possible during waking life.

**Table 4 brainsci-14-00496-t004:** Group comparisons for all instruments between frequent and non-frequent lucid dreamers within meditation groups.

	DFMs				Frequent Meditators				Non-Frequent Meditators			
	WFLDs	NWLDs			WFLDs	NWLDs			WFLDs	NWLDs		
	M (SD)	M (SD)	M Diff ^a^	*p* ^a^	M (SD)	M (SD)	M Diff ^a^	*p* ^a^	M (SD)	M (SD)	M Diff ^a^	*p* ^a^
FFMQ												
Observing	4.50 (0.07)	3.94 (0.6)	0.56	0.02 *	4.16 (0.8)	3.96 (0.7)	0.20	0.12	3.9 (0.7)	3.8 (0.8)	0.10	0.25
Describing	4.31 (0.8)	3.80 (0.6)	0.51	0.04 *	3.60 (1.04)	3.61 (0.71)	−0.01	0.47	3.38 (0.8)	3.11 (0.9)	0.27	0.23
Actaware	3.40 (0.9)	3.38 (0.9)	0.02	0.46	3.22 (0.9)	3.12 (0.8)	0.10	0.33	2.91 (0.9)	2.67 (0.8)	0.24	0.23
Nonjudge	4.27 (1.1)	3.74 (0.9)	0.53	0.14	3.93 (0.8)	3.89 (0.8)	0.04	0.46	3.54 (1.0)	3.35 (1.0)	0.19	0.24
Nonreact	4.02 (0.7)	3.47 (0.7)	0.55	0.04 *	3.6 (0.7)	3.42 (0.7)	0.26	0.14	3.32 (0.9)	3.00 (0.8)	0.32	0.23
TMS												
Decentering	2.59 (1.0)	2.53 (0.7)	0.07	0.46	2.54 (0.8)	2.32 (0.8)	0.22	0.14	1.73 (1.0)	1.60 (0.8)	0.14	0.25
Curiosity	2.41 (1.1)	2.27 (0.7)	0.14	0.42	2.30 (1.0)	2.15 (0.8)	0.14	0.26	1.86 (0.7)	1.65 (1.0)	0.15	0.24
RMMtm												
Transcend	4.52 (1.2)	3.80 (1.8)	0.72	0.18	4.01 (1.7)	3.56 (1.6)	0.55	0.14	2.78 (1.4)	2.45 (1.3)	0.33	0.23

Note: M Diff ^a^ = Mean difference between observed mean and mean constructed under the null hypothesis and R = 10,000 permutations; *p*
^a^ = *p*-value corrected with false discovery rate for each set of tests; * = *p* < 0.05; DFMs = daily frequent meditators; WFLDs = weekly lucid dreamers; NWLDs = non-weekly lucid dreamers; TMS = Toronto Mindfulness Scale; FFMQ = Five Facet Mindfulness Questionnaire; Actaware = Acting with Awareness; Nonjudge = Nonjudgment; Nonreact = Nonreactivity; RMMtm = Relaxation, Mindfulness and Meditation Tracker.

**Table 5 brainsci-14-00496-t005:** Multiple regression model for the relationship between lucid dreaming frequency and aspects of mindfulness, transcendence, meditation frequency, and meta-awareness, controlled for age.

Model	b	Bias ^a^	SE ^a^	95% BCa CI ^a^	
				Lower	Upper
Constant	−4.136	0.058	2.552	−9.346	0.727
Mindful Transcendence	1.228 *	−0.007	0.423	0.474	2.146
Decentering TMS	−0.303	−0.009	0.835	−1.921	1.367
Curiosity TMS	−0.441	0.004	0.792	−2.025	1.059
Meta-Awareness MAS	1.420 *	−0.006	0.633	0.227	2.684
Weekly Meditation Freq	0.029	0.002	0.171	−0.305	0.369
Age	−0.055	−0.0003	0.029	−0.113	0.0003

Note. ^a^ = Bootstrapped with R = 10,000 replicates; * = significant coefficient; SE ^a^ = standard error of the unstandardized coefficient; Model summary: R^2^ = 0.098; Residual standard error: 7.350; F (6, 263) = 4.780; *p* = 0.0001; *n* = 270; Weekly Meditation Freq = Weekly meditation frequency.

**Table 6 brainsci-14-00496-t006:** Regression models for complete mediation of the relationship of lucid dreaming occurrences in the previous 6-month period and weekly meditation frequency mediated by meta-awareness.

Outcome Variable	b	SE ^a^	95% BCa CI ^a^		R^2^	F^hc^
			Lower	Upper		
META-AWARENESS					0.124	9.836 ***
Intercept	4.661 *	0.221	4.215	5.085		
Meditation Freq	0.047 *	0.020	0.009	0.086		
Induction Freq	0.002	0.009	−0.017	0.018		
Dream Recall Freq	0.048 *	0.022	0.004	0.091		
Age	0.014 *	0.004	0.006	0.021		
LUCID DREAMS 6M					0.305	4.778 ***
Intercept	−17.849	20.459	−62.946	9.033		
Meditation Freq	0.702	1.366	−1.610	3.904		
Meta-Awareness	6.481 *	3.626	0.122	14.650		
Induction Freq	4.913 *	1.485	2.358	8.296		
Dream Recall Freq	0.305	0.939	−1.605	2.125		
Age	−0.357 *	0.201	−0.857	−0.037		

Note. ^a^ = Bootstrapped with R = 10,000 replicates; SE ^a^ = standard error of the unstandardized coefficient; * = significant coefficient; *** = *p* < 0.001; F^hc^ = robust estimate for standard error due to heteroscedasticity; *n* = 245; Lucid Dreams 6M = number of lucid dreams in the previous six-month period; Meditation Freq = Weekly meditation frequency; Induction Freq = Monthly frequency of induction of lucid dreams; Dream Recall Freq = Weekly dream recall frequency.

## Data Availability

The raw data supporting the conclusions of this article will be made available by the authors on request. The data are not publicly available due to privacy restrictions.

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
