# Peer review of "Frequent Lucid Dreaming Is Associated with Meditation Practice Styles, Meta-Awareness, and Trait Mindfulness"

_brainsci, 2024, doi:10.3390/brainsci14050496_

Round 1

Reviewer 1 Report

Comments and Suggestions for Authors

The article reports an extensive online survey (N=635), involving several measures of dreams, mindfulness, meditation, meta-awareness, for entangling relationships among them. As such, it is a useful contribution to the field, expanding on the previous findings and adding some novel observations. However, certain revisions are needed before the paper can be accepted for publication.

Major issues:

1.        The authors collected a lot of data (for the respondents, the questionnaire took about an hour to complete) and are presenting very extensive findings, with numerous cross-sectional comparisons, sub-group analyses, etc., so at the end the reader can’t really see the forest for the trees… I would suggest reconsidering how the findings are presented, and try to simplify this – e.g. more focus on the key aspects, try to depict them more visually (in tables, figures) and do not repeat the same results both in the text and tables (which currently happens on several occasions). 

2.        I found the divisions between monthly-frequent vs. weekly-frequent lucid dreamers, and especially weekly-frequent vs. daily-frequent meditators somewhat artificial and the categories by the definitions provided are not mutually exclusive (e.g. all weekly-frequent lucid dreamers will fit into the monthly-frequent lucid dreamers category). And why would daily-frequent meditators be required to meditate at least twice a day? Then someone meditating 5 mins in the morning and 5 mins in the evening will be a daily-frequent meditator, but someone meditating for an hour each day (but once) will count only as a ‘weekly-frequent’ meditator…  And this results in extremely small samples of ‘daily-frequent’ meditators (like 0.4% in UT Austin student sample). Considering the scales used, it might be better to simply consider frequency of lucid dreams and meditation practice as ordinal variables in statistical analyses (and conversions to the metric frequencies are also questionable – e.g. why frequency of more than 1 dream per night converts to 9 dreams per week yet 33 dreams per month?). The notions are also not consistently retained (e.g. Table 4 lists simply ‘Frequent Meditator’).

3.        The Introduction section seemed to be somewhat fragmented, without a sufficiently clear progression of ideas. It might benefit from being more structured and/or reorganized. To some extent this also applies to Discussion.

Minor points:

1.        Line 11: “Lucid dreaming involves gaining higher-order awareness during dreaming” – sounds somewhat abstract (“higher-order awareness”)

2.        Lines 43-44: “induction of lucid dreams can be trained by a set of cognitive practices or enhanced by substances” – not only, see the systematic reviews (Stumbrys et al., 2012; Tan & Fan, 2023)

3.        Line 49: Would be useful to expand on Tibetan Dream and Sleep Yogas

4.        Lines 53-55: Transcendental Meditation specifically is a fairly new development by Maharishi Mahesh Yogi in 1950s and has not been there “for ages” (although it does have roots in ancient Hindu tradition)

5.        Line 58: What does ‘shifting’ mean in the context of meditation?

6.        Lines 66-67: “Mindfulness meditation can be practiced to develop various skills and traits, such as cultivating compassion through loving kindness (LK) meditation” – yes, it can, but certainly not this only skill

7.        Lines 138-141: “In the latest study by Geise and Smith (2023), the Transcendence subscale of the Relaxation, Mindfulness, and Meditation Experience Tracker was found to be a significant predictor of lucid dreaming frequency” – both the scale and the subscale need some explanation 

8.        Lines 163-164 (as well as 728-729): “there has not been any study investigating which meditation technique is associated with higher lucid dreaming frequency” – incorrect, there is in fact such research (García-Campayo et al., 2021), showing by the way similar findings as in the present study – the association between OM and lucid dreaming.

9.        Lines 168-169: “We evaluated individual differences between frequent meditators and non-meditators” – needs to be specified what differences were evaluated

10.  Lines 198 & 201: “Range = 30” (or “ = 20”) does not depict the range

11.  Lines 214-215: Why there was such a huge difference in time required to complete the survey?

12.  Lines 273-274: “which is a summative measurement of lucid dream frequency as opposed to a formative approach” – needs clarification

13.  Lines 292-294: Please list those 20 frameworks, otherwise this lacks clarity

14.  Lines 301: “how many frameworks they meditated on” – sounds a bit awkward… Perhaps – how many different techniques they used in their meditation practice?

15.  It is unclear why several measures of dispositional mindfulness were used?

16.  Line 483: Unclear what is meant by “0 = practiced hours per week > 50”

17.  Lines 513-516: “In addition, the numbers of lucid dreams in the last six-month period revealed insignificantly higher numbers of lucid dreams in frequent meditators (M = 33.30, SD = 87.69, Range = 0 – 720) compared to non-frequent meditators (M = 20.44, SD = 41.86, Range = 0 – 320) [p = 0.018” – well, that is significant!

18.  Tables 3 & 4: References are to “Frequent Medi.” and “Frequent Meditator” – but you had two different categories (see the major point 2 above). And it also quite confusing that in different places (e.g. different tables) somewhat different abbreviations for the same variable are used.

19.  Line 722: “OM meditation is postulated to evolve from FA meditation” – I am not sure about the accuracy of this claim. FA can often be used to focus attention before moving into OM, but essentially and historically these are two different paths of meditation (see e.g. Goleman, 1996).

20.  There are several inaccuracies with referencing, e.g.

a.        Should be Holecek (not Holocek)

b.        Dalai Lama and Varela (not Valera)

c.        Travis (1970) is inaccurate as he hasn’t published any paper by then

Additional references

García-Campayo, J., Moyano, N., Modrego-Alarcón, M., Herrera-Mercadal, P., Puebla-Guedea, M., Campos, D., & Gascón, S. (2021). Validation of the Spanish Version of the Lucidity and Consciousness in Dreams Scale. Frontiers in Psychology12, 742438.

Goleman, D. (1996). The meditative mind: The varieties of meditative experience. Thorsons.

Tan, S., & Fan, J. (2023). A systematic review of new empirical data on lucid dream induction techniques. Journal of Sleep Research32(3), e13786.

Comments on the Quality of English Language

The quality of English is appropriate 

Reviewer 2 Report

Comments and Suggestions for Authors

The paper is well written and interesting. The text is well written, but it is very long. It is difficult to follow, I suggest the development of a scheme of the study, or a graphic abstract, which will facilitate an easier understanding of the steps that have been taken.

Author Response

R2:"The paper is well written and interesting. The text is well written, but it is very long. It is difficult to follow, I suggest the development of a scheme of the study, or a graphic abstract, which will facilitate an easier understanding of the steps that have been taken."

We thank the reviewer for the positive feedback. In the revised manuscript we have shortened the manuscript length as well as restructured the introduction and discussion in order to facilitate reader comprehension.

Reviewer 3 Report

Comments and Suggestions for Authors

Thanks to the authors for sharing their manuscript. My comments are insignificant and mostly relate to the structuring and design of the manuscript:

1.     The abstract looks disproportionate in structure, the background occupies almost half of its volume, the method is not described (participants and instruments). I would restructure the abstract.

2.     The sample of the study is not very large. Given that regression analysis and mediation analysis methods were used to analyze the data, I think it is logical to describe the calculations of sample size and statistical power.

3.     I would structure and systematize the limitations and future directions (maybe even shorten this section). I would also consider highlighting the subsections in the introduction and discussion. Since these parts of the manuscript are quite voluminous, it is easier for readers to perceive them if the authors structure them.

4.     Regarding the size of the manuscript and sources: how justified are the references to 120 sources in a manuscript describing an empirical study?

5.     Comment on the design: the authors use different versions of the numerical designations (for example, 0.22 and .22). In addition, the manuscript combines different fonts.

Sincerely yours,

the reviewer.

Author Response

R3:"Thanks to the authors for sharing their manuscript. My comments are insignificant and mostly relate to the structuring and design of the manuscript:"

We thank the reviewer for the positive feedback.

  1. The abstract looks disproportionate in structure, the background occupies almost half of its volume, the method is not described (participants and instruments). I would restructure the abstract.

We have added details for the sample size and study design (online survey) to the abstract. While we agree that substantial background is given, we think this information is crucial to orient readers to the current study.

  1. The sample of the study is not very large. Given that regression analysis and mediation analysis methods were used to analyze the data, I think it is logical to describe the calculations of sample size and statistical power.

As there was no previous study on this topic, it was not possible to conduct a power analysis prior to data collection. We aimed to collect as much data as possible given institutional and financial constraints. Nevertheless, sample sizes of 200 to 300 respondents are generally considered adequate to provide an acceptable margin of error in the social sciences (approximately 5%) before the point of diminishing returns (e.g., Kevin Lyons, Lipman Hearne).

  1. I would structure and systematize the limitations and future directions (maybe even shorten this section). I would also consider highlighting the subsections in the introduction and discussion. Since these parts of the manuscript are quite voluminous, it is easier for readers to perceive them if the authors structure them.

We thank the reviewer for this suggestion. We have restructured and shortened the limitations and future direction section as we agree it was too long in the previous draft.

  1. Regarding the size of the manuscript and sources: how justified are the references to 120 sources in a manuscript describing an empirical study?

We agree and have cut the references down to below 100.

  1. Comment on the design: the authors use different versions of the numerical designations (for example, 0.22 and .22). In addition, the manuscript combines different fonts.

In the revised manuscript we have ensured that only one font is used and that all numerical designations are consistent.